# Hormonal Regulation of Early Fruit Development in European Pear (*Pyrus communis* cv. 'Conference')

**Muriel Quinet** [1],*, **Christel Buyens** [1], **Petre I. Dobrev** [2], **Václav Motyka** [2] and **Anne-Laure Jacquemart** [1]

1 Earth and Life Institute-Agronomy, Genetics, Reproduction and Populations Research Group, Université catholique de Louvain, Croix du Sud 2, Box L7 05 14, 1348 Louvain-la-Neuve, Belgium; christel.buyens@uclouvain.be (C.B.); anne-laure.jacquemart@uclouvain.be (A.-L.J.)
2 The Czech Academy of Sciences, Institute of Experimental Botany, Laboratory of Hormonal Regulations in Plants, Rozvojová 263, CZ-16502 Prague 6, Czech Republic; Dobrev@ueb.cas.cz (P.I.D.); Motyka@ueb.cas.cz (V.M.)
* Correspondence: muriel.quinet@uclouvain.be; Tel.: +32-10-473443

**Abstract:** European pear requires inter-cultivar cross-pollination by insects to develop fertilized fruits. However, some European pear cultivars such as 'Conference' naturally produce parthenocarpic seedless fruits. To better understand the hormonal regulation of fruit set and early fruit development in this European pear cultivar, the phytohormone and polyamine profiles in 'Conference' flowers and fruits resulting from both fertilization and parthenocarpic processes were analyzed. The expression of genes involved in phytohormone metabolism and signaling were also investigated. Phytohormone profiles differed more at flower stage 3 days after treatment than in 15 day- and 30-day-old fruits in response to fertilization and parthenocarpy. An increase in auxins, abscisic acid, ethylene precursor, and spermine, and a decrease in putrescine were recorded in the fertilized flowers as compared to the parthenocarpic flowers. Fertilization also upregulated genes involved in gibberellin synthesis and down-regulated genes involved in gibberellin catabolism although the total gibberellin content was not modified. Moreover, exogenous gibberellin (GA3, GA4/7) and cytokinin (6BA) applications did not increase parthenocarpic induction in 'Conference' as observed in other European and Asian pear cultivars. We hypothesize that the intrinsic parthenocarpy of 'Conference' could be related to a high gibberellin level in the flowers explaining why exogenous gibberellin application did not increase parthenocarpy as observed in other pear cultivars and species.

**Keywords:** fruit set; parthenocarpy; phytohormones; pollination; polyamines; transcriptomic

## 1. Introduction

Fruit set and fruit development are complex developmental processes which need the coordination of different phytohormones [1–7]. For most plants, fruit set and development are triggered by pollination and fertilization [8,9]. Growth factors by which pollen influence fruit set include auxins and gibberellins (GAs) [9–11]. Gibberellins produced by the pollen may increase auxin production in the ovary, which in turn may act as a signal for fruit set and subsequent activation of cell division [9,12]. Active fruit growth by pericarp cell division and elongation is due to the synthesis of auxins in the developing seeds and of GAs in the pericarp [13]. In many species, auxin and cytokinin (CK) levels in the seeds increase during seed development until maturity [7,10]. Seeds communicate through hormones to the surrounding tissues to promote fruit growth [7,11]. Fruit set relies thus on the concerted action of auxins and/or GAs and/or CKs depending on the species [7,13]. A GA-auxin crosstalk is involved in early fruit development while each phytohormone seems to also play a specific

role on its own [7,8,12]. Cytokinins, brassinosteroids and polyamines (PAs) might also have a role in fruit set [7,8,10]. However, while auxin, GA and CK levels are increasing at fruit set, concentrations of abscisic acid (ABA) and ethylene decrease after pollination [7,12]. Once the fruit is initiated, fruit growth is due primarily to cell division and then continues, mostly by cell expansion, until the fruit reaches its final size [9]. Cell division and cell elongation activity are coordinated by a delicate balance between GAs and auxins during fruit development even if other phytohormones such as CKs and brassinosteroids may be involved [7,12–14]. Abscisic acid has also been associated with the expansion phase in some species [7]. Fruit maturation is the developmental point where the fruit has reached the competence to ripen; auxins and CKs appear to be key regulators of this step [7]. Once the seeds reach maturity, fruits become ready to undergo ripening, and during this period there is a major switch in the relative hormone levels of the fruit, involving an overall decrease in auxins, GAs, and CKs and a simultaneous increase in ABA and ethylene [1,2,7,10]. Some fruit types (referred as climacteric fruits) have a strong requirement for ethylene to ripen while others (referred as non-climacteric fruits) mainly depend on ABA to ripen [3,7]. However, ABA seems also to be important for the ripening of climacteric fruits [3,7]. In some plants such as tomato, peach and pear, auxins have also been reported to have some crosstalk with ethylene during ripening [3,7]. Moreover, brassinosteroids (BRs), jasmonates (JAs) and PAs were shown to influence fruit ripening [3,10].

Given their role in fruit set and development, induction of extrinsic parthenocarpy by phytohormone spraying is a common practice to induce seedless fruits in different fruit species. Parthenocarpy is characterized by the development of fruits without pollination and fertilization of the ovules resulting in seedless fruits [15,16]. Phytohormones such as GAs, CKs and auxins promote extrinsic parthenocarpy in several species [4,11–13,17,18]. As most fruit species of the Rosaceae family, pear (*Pyrus* sp.) exhibits natural self-incompatibility and requires inter-cultivar cross-pollination by insects to develop fertilized fruits [19,20]. To overcome self-incompatibility in pear, the production of fruits without fertilization through extrinsic parthenocarpy has been investigated [16,21–24]. Plant growth regulators such as GAs (GA3, GA4, GA7), melatonin and CKs (CPPU: $N$-(2-chloro-4-pyridyl)-$N'$-phenylurea; 6BA: $N^6$-benzyladenine) induced extrinsic parthenocarpy in Asian and European pears while auxins were not effective at inducing parthenocarpy in these species [20–23]. The effectiveness of the different phytohormones depends on the pear species (*Pyrus communis, P. pyrifolia, P. ussuriensis* and *P. bretschneideri*) and on the cultivars [16,22,23]. Usually, parthenocarpic pear fruit have a smaller size than pollinated fruit [16,22,23]. However, the precise role of phytohormones in regulating parthenocarpic fruit set and development is still poorly understood. Parthenocarpic fruit development has been mainly investigated in tomato (*Solanum lycopersicum* L.), and auxins, and GAs played a particular role in parthenocarpic fruit initiation in this species [10–12,18]. Overexpression of the auxin receptor TIR1 induced parthenocarpic fruit in tomato [18]. Likewise, mutants or transgenic plants with decreased expression of genes coding for auxin signal repressors such as auxin response factors (ARFs) or auxins/indole-3-acetic acid (Aux/IAA) proteins produced parthenocarpic fruit [12,18,25,26]. Parthenocarpic fruit development is also associated with higher expression of GA biosynthetic genes (such as *GA20ox*), increase of GA content or silencing of GA catabolic genes (*GA2ox*) and GA negative regulators (*DELLA*) in tomato [18,25,27–30]. The levels of PAs were also modified in parthenocarpic tomato mutants, indicating a possible role for these growth regulators in parthenocarpy induction [10]. In the Chinese white pear (*Pyrus bretschneideri*), induction of parthenocarpic fruits by GA4/7 treatment affected GA, IAA and ABA concentrations and metabolism, increasing the GA and IAA levels and decreasing the ABA levels [4]. In European pear (*Pyrus communis*), it has been proved that melatonin increases the content of bioactive GAs by the upregulation of *PbGA20ox* and downregulation of *PbGA2ox* [21].

However, unlike Asian pear species, some European pear cultivars naturally produce intrinsic parthenocarpic fruits independently of phytohormone spraying [15,16,22–24]. Such intrinsic parthenocarpy has been demonstrated in the 'Conference' cultivar but extrinsic parthenocarpy induction by spraying phytohormones, mainly GAs, remains a common practice in 'Conference'

orchards [16]. We previously showed that although intrinsic parthenocarpy occurred in 'Conference', hormonal treatments may increase the parthenocarpic fruit set under adverse environmental conditions [16]. Moreover, treatment with the GA biosynthesis inhibitor paclobutrazol did not completely prevent parthenocarpic fruit initiation, suggesting that endogenous GAs do not act alone to induce parthenocarpy in 'Conference' [16]. To improve our understanding of the hormonal regulation of the fruit set and early fruit development in European pear, we analyzed the complete phytohormone and PA profiles in 'Conference' fruits resulting from fertilization and parthenocarpic processes. We also investigated the expression of genes involved in phytohormone and PA metabolism and signaling. The aims of the study were: (1) to compare the fruit produced by fertilization to those by intrinsic parthenocarpic processes and to investigate the underlying modifications in phytohormone and PA profiles and metabolism during early fruit development, and (2) to investigate the effects of extrinsic parthenocarpy induced by GA (GA3 and GA4/7) and CK (6BA) on fruit production as well as on phytohormone and PA profiles and metabolism.

## 2. Materials and Methods

### 2.1. Sites and Plant Material

This study was carried out on European pear (*Pyrus communis* cv. 'Conference') trees in a 0.6 ha orchard of the CEF (Centre Fruitier Wallon) in Wasseiges, Belgium (50°37′58″ N; 5°0′57″ E). 'Conference' trees on Quince C were planted at 3.4 × 1.3 m (2262 trees/ha) in 1995. European pear cultivars 'Concorde', 'Triomphe de Vienne' and 'Doyenné du Comice' were present as pollinizer cultivars and were planted in the same row as the main cultivar at a mean ratio of 1 pollinizer tree to 16 'Conference' trees. All orchard management practices were identical to those for commercial production except that no thinning was performed on the analyzed trees. One honeybee hive was placed in the orchard during the flowering period as hive density recommendations ranged from 1 to 5 hives/ha for pear tree orchards [31].

### 2.2. Pollination and Hormonal Treatments

During 2 consecutive years, different pollination and phytohormone treatments were applied on three 'Conference' trees per treatment. Ten to 15 flower clusters on 2-year-old wood were selected per tree and covered with exclusion bags during flowering to prevent visits by pollinators and were followed during the growing period. The following treatments were applied: (1) flower emasculation (spontaneous intrinsic parthenocarpy); (2) self-pollination: incompatible hand pollination with 'Conference' pollen (induced intrinsic parthenocarpy); (3) cross-pollination: compatible hand pollination with 'Triomphe de Vienne' and 'Doyenné du Comice' pollen (fertilization); (4) phytohormone spraying with GAs (7.5 mg/L GA3 or 7.5 mg/L GA4/7), or CK (100 mg/L $N^6$-benzyladenine (6BA)) (extrinsic parthenocarpy). These treatments were compared with flower clusters freely pollinated by insects (open-pollination). The emasculation and hand pollination treatments were made to the bagged flower clusters. The cross-pollinations were made to emasculated flowers to avoid self-pollination. The phytohormone applications were made at full bloom, and phytohormone concentrations were selected based on previous results [16].

For each treatment, the number of flowers, number of initiated fruit and number of fruit at harvest were followed on 10–15 flower clusters per tree. At harvest, the resulting fruit were analyzed to determine their weight, size (maximum diameter), length, sugar content, and the presence of aborted and viable seeds. Sugar content was measured using a hand refractometer (Eclipse Handheld Refractometer; Bellingham & Stanley Ltd, Tunbridge Wells, UK). Over the 2 years, 100–140 fruits per treatment were measured at harvest.

Per treatment, 10 flowers were harvested 3 days after treatment (flowers at anthesis) and 10 fruits at 15 d (fruit size of 7–10 mm) and 30 d (fruit size of 12–15 mm) after treatment to analyze phytohormone and PA profiles and gene expression during fruit set and early fruit development. The harvested

flowers were composed of sepals, petals, stamens, and carpels, and the harvested young fruits included pericarp, developing seeds (if any), persistent sepals and remains of stamens.

## 2.3. Hormonal Quantification

Concentrations of the endogenous PAs and phytohormones including CKs, auxins (IAA), GAs, salicylic acid (SA), JAs, ABA, ethylene precursor 1-aminocyclopropane-1-carboxylic acid (ACC), benzoic acid (BzA), and their metabolites were determined, from flowers 3 days after treatment, 15-day-old fruit and 30-day-old fruit, each in triplicate. Flowers were analyzed for the emasculation, cross-pollination and open-pollination treatments; fruits were analyzed for the emasculation, self-pollination, cross-pollination, open-pollination and GA3, GA4/7 and 6BA hormonal treatments.

Phytohormones were extracted with methanol/formic acid/water (15:1:4, by volume) from liquid nitrogen-frozen and homogenized tissues and were subsequently purified by using a dual-mode solid-phase method [32]. Two phytohormone fractions were obtained; fraction A contained the acidic and neutral hormones (IAA, GAs, SA, JAs, ABA, BzA) and fraction B contained the basic hormones (CKs, ACC). The hormonal analysis and quantification were performed by high performance liquid chromatography (HPLC) (Ultimate 3000, Dionex, Sunnyvale, CA, USA) coupled to a hybrid triple quadropole/linear ion trap mass spectrometer (3200 Q TRAP; Applied Biosystems, Foster City, CA, USA) using a multilevel calibration graph with stable isotope labelled internal standards [33]. For fraction A, 10 μL of sample were injected onto a Luna C18 column ($100 \times 2$ mm internal diameter, 3 μm particle size; Phenomenex) and the mobile phase consisted of 5 mM ammonium formate (pH3)/acetonitrile gradient from 10 to 50% acetonitrile over 15 min. For fraction B, 10 μL of sample were injected onto a Luna C18 column ($150 \times 2$ mm internal diameter, 3 μm particle size; Phenomenex) and the mobile phase consisted of a 5 mM ammonium acetate (pH4)/methanol gradient from 5 to 40% methanol over 20 min. The flow rate was 0.25 mL·min$^{-1}$.Free PAs were extracted twice with 4% HClO$_4$ (v/v) at 4 °C and derivatized by dansylation as described in [34]. Samples were re-suspended in methanol, filtered (Chromafil PES-45/15, 0.45 μm; Macherey-Nagel, Düren, Germany) and injected onto a Nucleodur C18 Pyramid column ($125 \times 4.6$ mm internal diameter, 5 μm particle size; Macherey-Nagel) and maintained at 40 °C. Analyses were performed by a Shimadzu HPLC system coupled to a RF-20A fluorescence detector (Shimadzu, 's-Hertogenbosch, The Netherlands) as described in [35] with an excitation wavelength of 340 nm and an emission wavelength of 510 nm. The mobile phase consisted of a water/acetonitrile gradient from 40 to 90% acetonitrile over 20 min followed by a water/acetonitrile gradient from 90 to 100% acetonitrile over 2 min and the flow was 1.0 mL·min$^{-1}$.

## 2.4. Gene Expression Analysis

Some genes involved in phytohormone and PA metabolism and response (Table 1) were analyzed in pear flowers 3 days after treatment and in 15-day-old and 30-day-old fruits. Flowers were analyzed for the emasculation, cross-pollination, and open-pollination treatments; fruits were analyzed for the emasculation, self-pollination, cross-pollination, open-pollination, and GA3 and GA4/7 hormonal treatments.

Total RNA was prepared from 150 mg of material using the RNeasy Plant Mini Kit (Qiagen, Venlo, The Netherlands) and DNase treatments were realized using RQ1 RNAse-free DNase (Promega, Leiden, The Netherlands) according to the manufacturer's instructions. The concentration and quality of RNA were checked using the NanoDrop ND-1000 (Isogen Life Science, De Meern, The Netherlands). Reverse transcription was performed with 500 ng of total RNA using the RevertAid H Minus First Strand cDNA Synthesis Kit (Fermentas, St. Leon-Rot, Germany) by following the manufacturer's instructions. At least three independent PCR amplifications were conducted for each gene using the primer pairs, annealing temperatures, and the number of cycles presented in Table 1. Expression differences were analysed by gel densitometry using ImageJ software and were expressed as relative values compared to actin expression (peak size of target gene/peak size of actin) [35]. Primers were

designed using Primer3 software. Most primers were designed from *Pyrus communis* sequences and when primers were designed from *Malus domestica* or other *Pyrus* species, the amplified fragments were sequenced to verify their identity based on sequence alignments.

**Table 1.** List of primers and amplification conditions used for semi-quantitative RT-PCR expression analysis.

| Gene Name | Gene Function | Genbank Accession | Primer Sequences | Tm | Cycles |
|---|---|---|---|---|---|
| *ACTIN* | actin | AF386514 | TGCCTATGTAGGGGATGAGG GCTCAGCAGTTGTGGTGAAA | 55 °C | 30 |
| | | | Auxin signaling | | |
| *PpARP1* | Auxin-repressed protein | KC422235 | AGGTAGCAACCTTGCCACCAAAT GATACCTAATCGTATTGCCATC | 55 °C | 30 |
| *PpARP2* | Auxin-repressed protein | KC422236 | AGCGGTAACGGTGGATCAGCTCC CATCTCATGGATGATTTCGATTGGA | 55 °C | 30 |
| | | | Gibberellin metabolism and signaling | | |
| *PcCPS* | copalyl diphosphate synthase | KC153028 | GTGGCGTTAGTGGAGGATGT TCTCATGCAACACAGCACAA | 55 °C | 35 |
| *PpKS* | ent-kaurene synthase | JF441169 | CGATTGTCCTTCCAGCTCTC GTGCAGCACTTTGCTCATGT | 55 °C | 35 |
| *MdKAO1* | ent-kaurene acid oxidase | KF437682 | CGCAGAAGGGCTTAACACTC CGGATTAGTGCGTTCCATCT | 58 °C | 35 |
| *MdKAO2* | ent-kaurene acid oxidase | NM_001328825 | CGCAGAAGGGCTTAACACTC CGGATTAGTGCGTTCCATCT | 55 °C | 35 |
| *GA20ox* | gibberellin 20-oxidase | HQ833589 | CGAAAGCATCGCAACTTGTA GTGTGCTCATCGCCCTCACTA | 55 °C | 37 |
| *GA2ox* | gibberellin 2-oxidase | XM_008347715 | TTGAGCAAAGGAATGTGCTG AGTGACGGCAGAGGTGCTAT | 55 °C | 37 |
| *GA3ox* | gibberellin 3-oxidase | JX308225 | GTGCATCAGCTGCCTTACAA CCAAGTAATTGGCCGGTAGA | 55 °C | 37 |
| *GID1a* | GA signal transduction factor | JF516247 | TGCCTATGATGATGGATGGA GCTCTTCGGGAACTTGACAG | 55 °C | 30 |
| *GID1b* | GA signal transduction factor | JN381497 | GGAAAGTCCCTGCCAACATA GCAGCCACTTTCTCGATTTC | 55 °C | 30 |
| *GID1c* | GA signal transduction factor | JN381498 | GCCTCCTCAACCGTGTTTAC GCAGCCACTTTCTCGATTTC | 55 °C | 30 |
| *GID1d* | GA signal transduction factor | JN381499 | GCCTATGATGATGGGTGGAC CTCTTCGGGAACTTGACAGC | 55 °C | 30 |
| *DELLA* | DELLA protein | JF304103 | CGTCCAGCAGAACAACTTCA AACTCGACGTGGATGGTTTC | 55 °C | 30 |
| | | | Cytokinin metabolism | | |
| *PbCKX3* | cytokinin oxidase/dehydrogenase | XM_009354391 | GTGACGATCCAGAAGCCATT CGAGACCGGTGTAAGTCCAT | 58 °C | 35 |
| *PbCKX5* | cytokinin oxidase/dehydrogenase | XM_009337490 | AGTGTTCAAGGGCATTTTGG CCGATGAAGGGCTGAAAATA | 53 °C | 35 |
| | | | Polyamine metabolism | | |
| *PbSAMS* | S-adenosylmethionine synthase | AF195233 | AACCAAGGTGGACAGGAGTG ACCCCTCTTCAGATCCAGGT | 55° | 30 |
| *PbSAMS2lc* | S-adenosylmethionine synthase | XM_009355137 | AACCAAGGTGGACAGGAGTG ACCCCTCTTCAGATCCAGGT | 55 °C | 28 |
| *PbSAMDC* | S-adenosylmethionine decarboxylase | JX624260 | GATCCTTCCAGATTCGGACA TGCTAGCAACATTGGAGTGC | 55 °C | 30 |
| *PbSAMDC2* | S-adenosylmethionine decarboxylase | KC414856 | TCTTCGAGCCTGGACTGTTT CAATTTTGTTGTGCCACAGG | 55 °C | 30 |
| *PbSAMDC3* | S-adenosylmethionine decarboxylase | KM670010 | TTTTCGAGCCGAGTGTCTTT CACAGCAAGGGAAATGGTTT | 55 °C | 30 |
| *PbSAMDCla* | S-adenosylmethionine decarboxylase | XM_009341666 | TCGCCTCCTTGACTTTGAGT GGCAGGAGATGAGAGTGAGG | 55 °C | 34 |
| *PbSAMDClb* | S-adenosylmethionine decarboxylase | XM_009375455 | CAGCTGAGTGCACCATTGTT AGCTACCTCCTCCGAAAAGC | 55 °C | 28 |
| *PbSAMDCle* | S-adenosylmethionine decarboxylase | XR_670231 | TCTTCGAGCCTGGACTGTTT CAATTTTGTTGTGCCACAGG | 55 °C | 33 |
| *PbODC1* | ornithine decarboxylase | KP144199 | CCCAAATGTTCCTTGTTGCT AAAGCTGTTTCGGCAAAGAA | 58 °C | 35 |
| *PbODC1l* | ornithine decarboxylase-like | XM_009336895 | TTGCCGAAACAGCTTTCACATTGGT GCCGTTAAAGTTGGTTCCAGCAG | 58 °C | 35 |
| *PbODC2l* | ornithine decarboxylase-like | XM_009336872 | TTGCCGAAACGGCTTTCACAATGGT CCATTAAAGTTGGTTCCAATGG | 58 °C | 35 |
| *Spmsl1* | spermine synthase-like | XM_009380504 | CGGTTGCTTGAGTAGCAAAT GATCGCCATAGTAAAATTTCC | 58 °C | 35 |
| *Spmsl2* | spermine synthase-like | XM_009353722 | CGGTTGCTTGAGTAGCAGAT TAACATAATGTCGAATGGCT | 58 °C | 35 |

All statistical analyses were conducted in SAS Enterprise Guide 7.1, except for the principal component analyses (PCA) and the heatmaps, which were conducted in R version 3.4.2 [36]. The normality of the data was estimated using Shapiro–Wilk tests, and homoscedasticity was checked using Levene's tests. The data were transformed when required to ensure normal distributions. The data are presented as means ± standard errors unless indicated otherwise.

Fruit parameters were compared among treatments using one-way analyses of variance (ANOVA) and phytohormone concentrations were compared among treatments using two-way ANOVAs. Post-hoc analyses were performed using SNK tests to investigate the differences among treatments and flower and fruit stages. Complete phytohormone and PA profiles in fertilized and parthenocarpic fruits were visualized using PCA analysis. Phytohormone and PA profiles of parthenocarpic fruits treated or not with hormones (GA3, GA4/7 and 6BA) were visualized using a heatmap. Gene expressions were also visualized using heatmaps.

## 3. Results

### 3.1. Differences Between Fertilized and Intrinsic Parthenocarpic Fruits

3.1.1. Effects of Fertilization and Intrinsic Parthenocarpy on Fruit Parameters

We compared fruit set and fruit development of 'Conference' fruits resulting from intrinsic parthenocarpy (bagged emasculated flowers and bagged self-pollinated flowers), open-pollination and compatible hand-pollination (cross-pollination) (Table 2). Emasculation decreased the initial fruit set as compared to open-pollination but the fruit set at harvest was similar whatever the treatment (Table 2). As expected, the parthenocarpic fruits did not contain normal seeds and contained fewer aborted seeds than the open-pollinated and cross-pollinated fruits. However, the seed set was low whatever the treatment. There were no differences in fruit weight but parthenocarpic fruit were more elongated than the cross-pollinated fruit. Fruit size was also lower in parthenocarpic fruits than in open-pollinated and cross-pollinated fruits. The sugar content was about 12% in both parthenocarpic and fertilized fruits.

**Table 2.** Effects of fertilization and intrinsic parthenocarpy on fruit set and fruit parameters in 'Conference' pear.

| Parameter | Initial Fruit Set (%) | Fruit Set at Harvest (%) | Fruit Weight (g) | Fruit Length (mm) | Fruit Size (mm) | Normal Seeds per Fruit | Aborted Seeds per Fruit |
|---|---|---|---|---|---|---|---|
| open-pollination | 86.8 ± 5.2a | 18.2 ± 6.1a | 138.6 ± 6.3a | 107.9 ± 2.1a | 57.8 ± 1.0a | 0.25 ± 0.09b | 1.95 ± 0.34a |
| emasculation | 56.7 ± 9.9b | 13.3 ± 2.1a | 120.3 ± 5.7a | 107.5 ± 2.5a | 51.4 ± 1.1b | 0.00 ± 0.00b | 0.97 ± 0.38b |
| self-pollination | 80.8 ± 4.4ab | 16.8 ± 6.1a | 118.7 ± 6.0a | 108.9 ± 2.2a | 51.1 ± 0.8b | 0.00 ± 0.00b | 0.38 ± 0.22b |
| cross-pollination | 62.6 ± 6.8ab | 18.5 ± 2.9a | 131.3 ± 5.5a | 99.8 ± 1.9b | 55.5 ± 0.9a | 1.91 ± 0.31a | 2.58 ± 0.36a |
| ANOVA | $F_{3,25} = 3.59$ $P = 0.0277$ | $F_{3,25} = 0.33$ $P = 0.8048$ | $F_{3,191} = 2.43$ $P = 0.0669$ | $F_{3,191} = 4.26$ $P = 0.0061$ | $F_{3,191} = 5.85$ $P = 0.0008$ | $F_{3,103.9} = 13.28$ $P < 0.0001$ | $F_{3,86.7} = 17.67$ $P < 0.0001$ |

3.1.2. Effects of Fertilization and Intrinsic Parthenocarpy on Phytohormonal Profile

The phytohormonal profiles changed in both fertilized and parthenocarpic fruits during early fruit development (Figures 1–3). Axis 1 of the PCA clearly discriminated the flowers and the fruits by their phytohormone profile and explained 32.56% of the variance (Figure 3a). The concentrations of ABA, IAA, BzA, CKs (bioactive CKs, CK-*N*-glucosides, CK-*O*-glucosides and CK phosphates), ACC, spermidine, and spermine were higher in the flowers than in the fruits whatever the treatment (Figures 1 and 2, Table 3). In contrast, the highest concentration of JAs was observed in the 30-day-old fruit (Figure 2e), and the highest concentration of putrescine was observed in the 15-day-old fruit (Figure 2h). The concentration of bioactive CKs (Figure 2a), CK-*O*-glucosides (Figure 2c), CK phosphates (Figure 2d), and spermidine (Figure 2f) progressively decreased with the age of the fruit.

Axis 2 of the PCA explained 11.78% of the variance and mainly separated the open-pollinated and emasculated flowers from the cross-pollinated flowers (Figure 3b). Differences among treatments were indeed mainly observed in the flowers (Figures 1 and 2). The concentration of ABA, IAA, bioactive CKs, ACC, and spermine (Figure 1a,b and Figure 2a,e,g) were higher in the cross-pollinated flowers than in the open-pollinated and/or emasculated flowers while the concentration of putrescine was higher in the open-pollinated flowers than in the cross-pollinated ones (Figure 2h). In the fruit, the concentrations of SA and putrescine in 15-day-old fruit were higher for the emasculation than for the self-pollination and open-pollination treatments (Figures 1c and 2h), while JA concentration in 30-day-old fruit was higher for open-pollination and cross-pollination than for self-pollination treatments (Figure 1e).

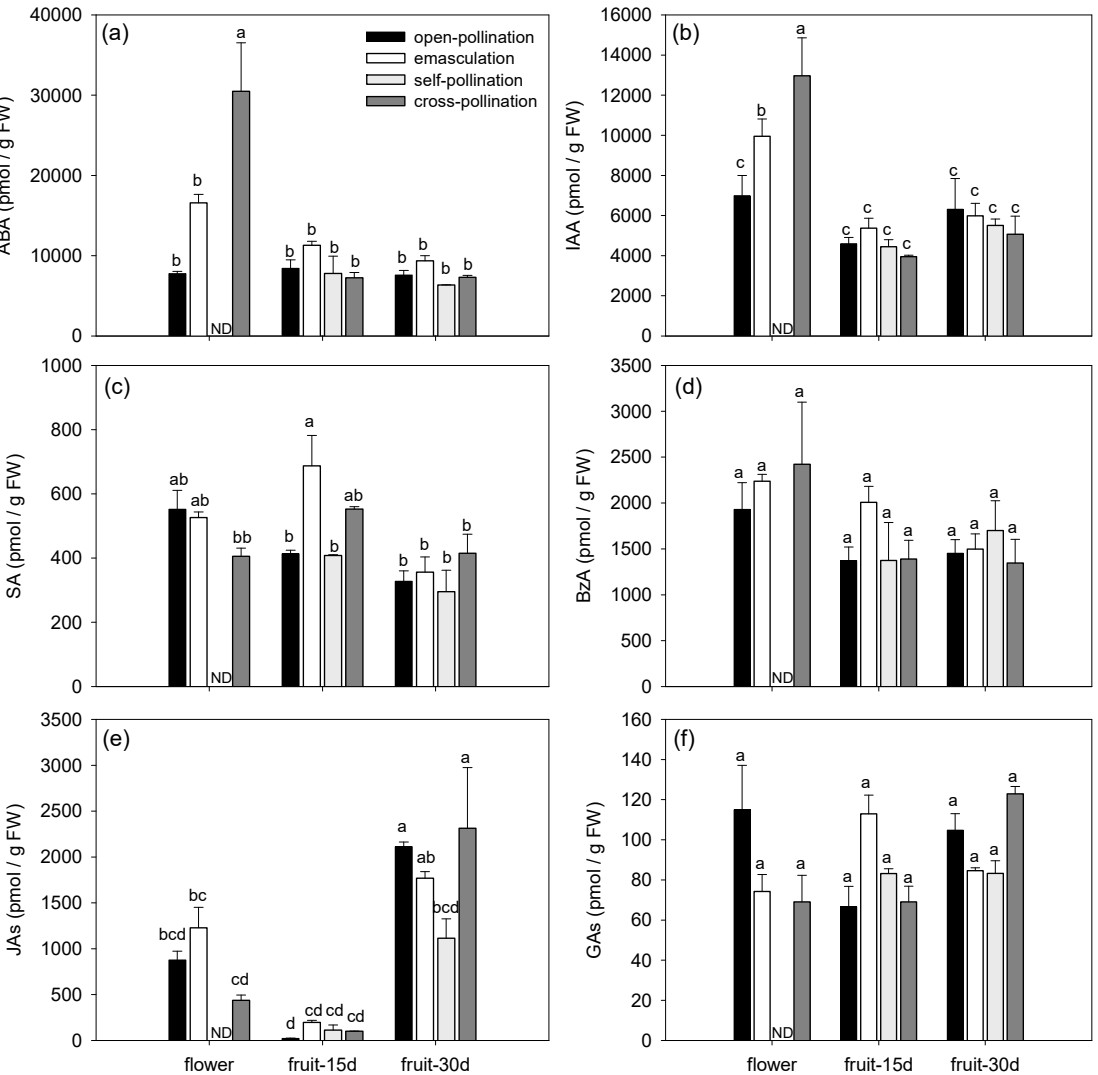

**Figure 1.** Endogenous phytohormone profile during early fruit development of fertilized and intrinsic parthenocarpic 'Conference' pear fruits. (**a**) Total abscisic acid (ABA), (**b**) total auxins (IAA and its derivatives), (**c**) salicylic acid (SA), (**d**) benzoic acid, (**e**) total jasmonates (JAs), and (**f**) total gibberellins (GAs).

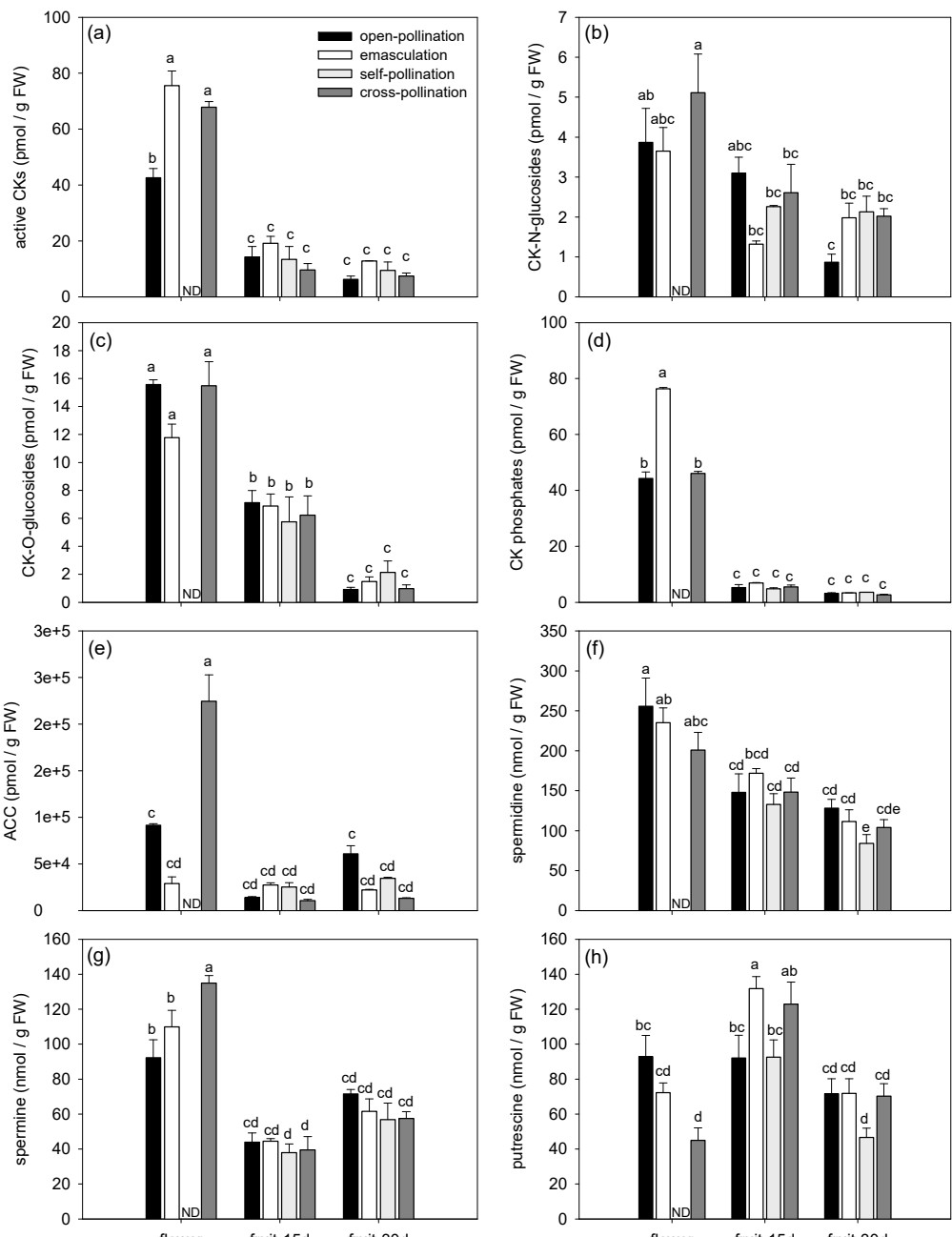

**Figure 2.** Endogenous phytohormone and polyamine profiles during early fruit development of fertilized and intrinsic parthenocarpic 'Conference' pear fruits. (**a**) bioactive cytokinins (CKs), (**b**) CK-*N*-glucosides, (**c**) CK-*O*-glucosides, (**d**) CK phosphates, (**e**) ethylene precursor ACC, (**f**) spermidine, (**g**) spermine, and (**h**) putrescine.

Regarding phytohormone metabolites, most metabolites grouped together as observed for cytokinin metabolites that were mainly present in flowers or ABA and IAA metabolites mainly present in cross-pollinated flowers (Figure 3a). However for some phytohormones, divergences among metabolites were detected (Figure 3a). For example, among JAs, the highest concentration of JA amino acid conjugate JA-isoleucine (JA-Ileu) was observed in the flowers while JA precursor *cis*-(+)-12-oxo-phytodienoic acid (*cis*OPDA) was mainly found in the 30-day-old fruit. Regarding the GAs, GA7 correlated with the CKs and was mainly present in the flowers, GA8 was found in high concentration in the emasculated flowers and GA19 was more abundant in 30-day-old fruit.

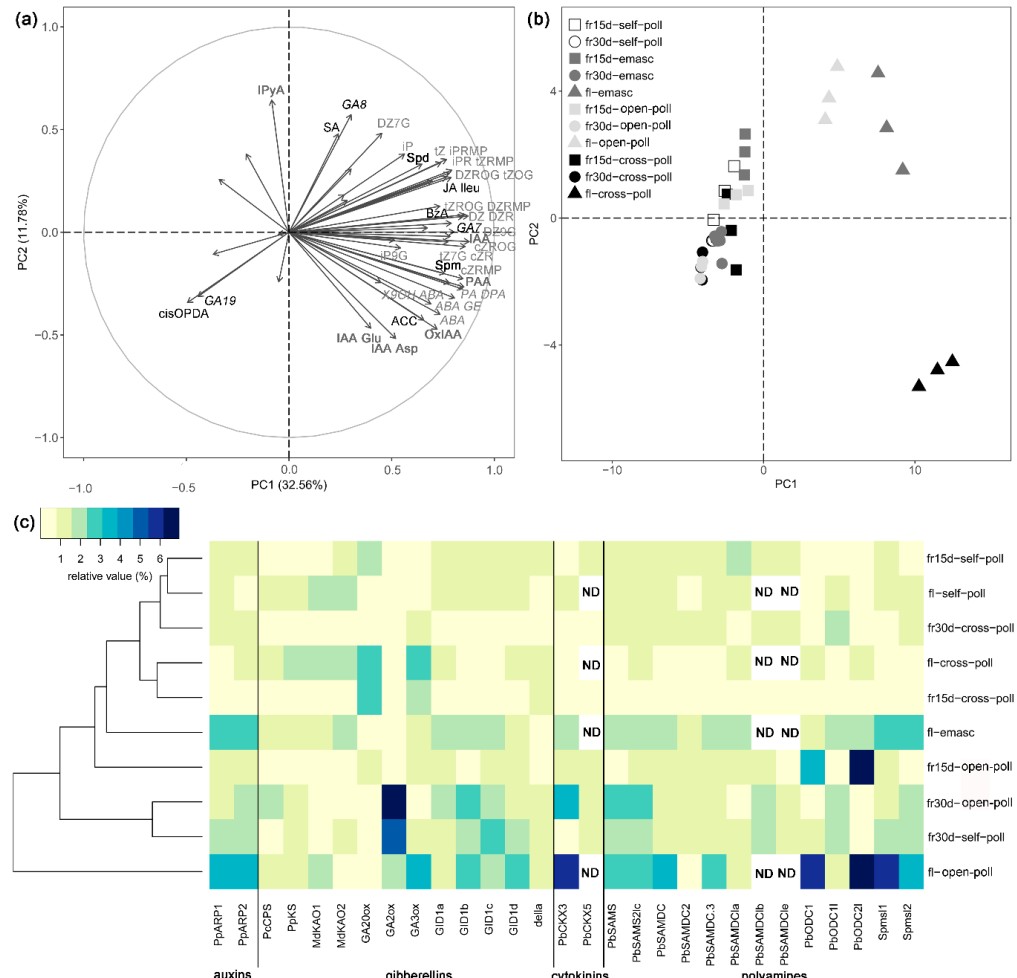

**Figure 3.** Effects of fertilization and intrinsic parthenocarpy on hormonal regulation during early fruit development of 'Conference' pear fruits. (**a-b**) Principal Component Analysis (PCA) of phytohormone and polyamine profiles in flowers and young 'Conference' fruits after different pollination and emasculation treatments; (**a**) variable graph and (**b**) individual graph; only significant parameters were shown (P< 0.05) in (**a**). (**c**) Heatmap of relative transcript levels of genes involved in phytohormone and polyamine metabolism and signaling in 'Conference' flowers and young fruits after different pollination and emasculation treatments. Abbreviations: Flowers (fl), 15-day-old fruits (fr15d) and 30-day-old fruits (fr30d) after emasculation (emasc), self-pollination (self-poll), cross-pollination (cross-poll) and open-pollination (open-poll). Auxins metabolites (IAA-Asp, indole-3-acetic acid aspartate; IAA-Glu, indole-3-acetic acid glutamate; OxIAA, oxo-indole-3-acetic acid; PAA, phenylacetic acid; IPyA, indole-3-pyruvic acid (IAA precursor), bold dark grey). Abscisic acid metabolites (ABA, abscisic acid; ABA-GE, ABA-glucose ester; PA, phaseic acid; DPA, dihydrophaseic acid; 9OH-ABA, 9-hydroxy-ABA; italic grey). Gibberellins (GA8, gibberellic acid 8; GA19, gibberellic acid 19; black italic). Jasmonates (JA-Ileu, JA-isoleucine; *cis*OPDA (JA precursor); black). Cytokinin metabolites (tZ, trans-zeatin; tZ7G, trans-zeatin 7-glucoside; tZOG, trans-zeatin O-glucoside; tZROG, trans-zeatin 9-riboside -O-glucoside; tZRMP, trans-zeatin 9-riboside-5′-monophosphate; cZR, cis-zeatin 9-riboside; cZOG, cis-zeatin O-glucoside; cZROG, cis-zeatin 9-riboside-O-glucoside; cZRMP, cis-zeatin 9-riboside-5′-monophosphate; DZ, dihydrozeatin; DZR, dihydrozeatin 9-riboside; DZ9G, dihydrozeatin 9-glucoside; DZROG, dihydrozeatin 9-riboside-O-glucoside; iP, $N^6$-($\Delta^2$-isopentenyl)adenine, iPR, $N^6$-($\Delta^2$-isopentenyl)adenosine; iP9G, $N^6$-($\Delta^2$-isopentenyl)adenine 9-glucoside; iPRMP, $N^6$-($\Delta^2$-isopentenyl)adenosine-5′-monophosphate; grey). SA, salicylic acid; BzA, Benzoic acid; PAA, Phenylacetic acid; ACC, 1-aminocyclopropane-1-carboxylic acid (ethylene precursor). Genes are annotated as described in Table 1.

**Table 3.** Statistical results (ANOVA2) of endogenous phytohormone and PA profiles during early fruit development of fertilized and intrinsic parthenocarpic 'Conference' pear fruits.

| Parameter | Treatments | Flower and Fruit Stages | Interaction |
|---|---|---|---|
| ABA | $F_{3,11} = 6.60$, $P = 0.0082$ | $F_{2,11} = 23.61$, $P = 0.0001$ | $F_{5,11} = 9.66$, $P = 0.0010$ |
| IAA | $F_{3,11} = 1.28$, $P = 0.3296$ | $F_{2,11} = 27.78$, $P < 0.0001$ | $F_{5,11} = 3.85$, $P = 0.0292$ |
| SA | $F_{3,11} = 4.53$, $P = 0.0266$ | $F_{2,11} = 12.92$, $P = 0.0013$ | $F_{5,11} = 3.75$, $P = 0.0316$ |
| BzA | $F_{3,11} = 0.58$, $P = 0.6396$ | $F_{2,11} = 5.00$, $P = 0.0286$ | $F_{5,11} = 0.68$, $P = 0.6490$ |
| JAs | $F_{3,11} = 2.34$, $P = 0.1300$ | $F_{2,11} = 59.83$, $P < 0.0001$ | $F_{5,11} = 3.19$, $P = 0.0504$ |
| GAs | $F_{3,11} = 0.88$, $P = 0.4810$ | $F_{2,11} = 3.05$, $P = 0.0885$ | $F_{5,11} = 6.73$, $P = 0.0042$ |
| bioactive CKs | $F_{3,11} = 11.93$, $P = 0.0009$ | $F_{2,11} = 280.64$, $P < 0.0001$ | $F_{5,11} = 7.35$, $P = 0.0030$ |
| CK-*N*-glucosides | $F_{3,11} = 1.71$, $P = 0.2231$ | $F_{2,11} = 18.96$, $P = 0.0003$ | $F_{5,11} = 1.84$, $P = 0.1859$ |
| CK-*O*-glucosides | $F_{3,11} = 0.70$, $P = 0.5710$ | $F_{2,11} = 127.94$, $P < 0.0001$ | $F_{5,11} = 1.81$, $P = 0.1922$ |
| CK phosphates | $F_{3,11} = 116.37$, $P < 0.0001$ | $F_{2,11} = 3757.22$, $P < 0.0001$ | $F_{5,11} = 117.34$, $P < 0.0001$ |
| ACC | $F_{3,11} = 18.47$, $P = 0.0001$ | $F_{2,11} = 95.31$, $P < 0.0001$ | $F_{5,11} = 38.02$, $P < 0.0001$ |
| spermidine | $F_{3,24} = 2.24$, $P = 0.1098$ | $F_{2,24} = 33.11$, $P < .0001$ | $F_{5,24} = 0.62$, $P = 0.6867$ |
| spermine | $F_{3,24} = 0.99$, $P = 0.4151$ | $F_{2,24} = 78.35$, $P < 0.0001$ | $F_{5,24} = 3.73$, $P = 0.0122$ |
| putrescine | $F_{3,24} = 4.26$, $P = 0.0151$ | $F_{2,24} = 28.62$, $P < 0.0001$ | $F_{5,24} = 3.98$, $P = 0.0090$ |

### 3.1.3. Effects of Fertilization and Intrinsic Parthenocarpy on Expression of Genes Involved in Phytohormone Metabolism

Expression of genes involved in auxin, GA, CK, and PA metabolism or signaling were compared among fertilized and intrinsic parthenocarpic fruits during early fruit development (Figure 3c). Globally, gene expression was higher in open-pollinated and emasculated flowers than in the other conditions as observed for the *Auxin Repressed Protein* (*ARP*) genes. The *Cytokinin Oxidase/Dehydrogenase* (*CKX*) genes, which regulate CK levels, were mainly expressed in the open-pollinated flowers and 30-day-old fruits. Regarding GA metabolism, genes involved in the first steps of GA biosynthesis (*PcCPS*, *PpKS*, *MdKAO1*, *MdKAO2*) showed a higher expression in the flowers than in the fruits. The highest *GA20ox* transcript level was observed in the fertilized flowers and young fruits, and *GA3ox* which is involved in the synthesis of bioactive GAs was mainly expressed in the fertilized flowers. However, *GA2ox* which encodes an enzyme involved in GA inactivation was mainly expressed in the open-pollinated and self-pollinated 30-day-old fruits. Genes involved in GA signaling such as *GID* were less expressed in fertilized flowers and fruits than in the other conditions, and the expression of *DELLA* was similar whatever the treatment. Genes involved in S-adenosyl methionine (SAM) and PA synthesis were also affected. Genes coding for SAM Synthase (*PbSAMS*, *PbSAMS2lc*) were mainly expressed in the open-pollinated flowers and 30-day-old fruits. The transcript levels of *SAM Decarboxylase* (*PbSAMDC*, *PbSAMDC2*, *PbSAMDC3*, *PbSAMSDCla*, *PbSAMSDClb*, *PbSAMSDCle*) were globally more expressed in the open-pollinated and emasculated flowers although their expression varied depending on the gene. The *Ornithine Decarboxylase* genes (*PbODC* and *PbODC2l*) which are required for putrescine synthesis were mainly expressed in the open-pollinated flowers and 15-day-old fruit, and the highest expression of *Spermidine synthase* genes (*SPmSl1*, *Spmsl2*) was observed in the open-pollinated and emasculated flowers.

### 3.2. Differences Between Intrinsic and Extrinsic Parthenocarpic Fruits

### 3.2.1. Effects of Extrinsic Parthenocarpy on Fruit Parameters

Extrinsic parthenocarpy induced by GAs (GA3 and GA4/7) or CK (6BA) treatment was compared to intrinsic parthenocarpy (bagged self-pollinated flowers) and open-pollination (Table 4). Neither the initial fruit set nor fruit set at harvest were affected by the hormonal treatments. Both intrinsic and extrinsic parthenocarpy decreased fruit weight, fruit size and seed production as compared to open-pollinated fruits but did not impact fruit length. As a result, parthenocarpic fruits were more elongated than open-pollinated fruits (length to size ratio of $2.23 \pm 0.03$ vs. $1.98 \pm 0.03$, $F_{4,344} = 12.81$, $P < 0.0001$). Regarding hormonal treatments, fruit size was smaller in 6BA-induced fruits than in GA3-induced fruits. Moreover, the fruit sugar content was reduced after hormonal

treatment (10.7–11.5 °Brix) compared to open-pollinated and intrinsic parthenocarpic fruits (12.4 °Brix, $F_{4,195}$ = 7.14, $P < 0.0001$).

**Table 4.** Effect of extrinsic parthenocarpy on fruit set and fruit parameters in 'Conference' pear.

| Parameter | Initial Fruit Set (%) | Fruit Set at Harvest (%) | Fruit Weight (g) | Fruit Length (mm) | Fruit Size (mm) | Normal Seeds per Fruit | Aborted Seeds per Fruit |
|---|---|---|---|---|---|---|---|
| open-pollination | 86.8 ± 5.2a | 18.2 ± 6.1a | 138.6 ± 6.3a | 107.9 ± 2.1a | 57.8 ± 1.0a | 0.25 ± 0.09b | 1.95 ± 0.34a |
| Intrinsic parthenocarpy | 80.8 ± 4.4ab | 16.8 ± 6.1a | 118.7 ± 6.0a | 108.9 ± 2.2a | 51.1 ± 0.8b | 0.00 ± 0.00b | 0.38 ± 0.22b |
| GA3 | 88.9 ± 4.1 a | 29.2 ± 3.1a | 133.6 ± 4.3 ab | 113.7 ± 1.8 a | 51.2 ± 0.8 b | 0.00 ± 0.00b | 0.14 ± 0.08 b |
| GA4/7 | 93.5 ± 2.0a | 24.6 ± 4.3 a | 117.5 ± 4.8 b | 111.6 ± 2.0 a | 49.9 ± 0.8 bc | 0.00 ± 0.00 b | 0.59 ± 0.23 b |
| 6BA | 79.2 ± 5.9 b | 17.4 ± 3.3 a | 120.8 ± 4.1 b | 107.7 ± 1.8 a | 48.1 ± 0.7 c | 0.03 ± 0.02 b | 0.14 ± 0.08 b |
| ANOVA | $F_{4,24}$ = 2.46 $P = 0.0726$ | $F_{4,24}$ = 1.34 $P = 0.2837$ | $F_{4,344}$ = 3.54 $P = 0.0076$ | $F_{4,344}$ = 1.86 $P = 0.1168$ | $F_{4,344}$ = 8.12 $P < 0.0001$ | $F_{4,85.17}$ = 2.76 $P = 0.0089$ | $F_{4,146.4}$ = 7.71 $P < 0.0001$ |

### 3.2.2. Effects of Extrinsic Parthenocarpy on Phytohormonal Profile

As observed in Figure 4a, the phytohormone and PA profiles varied according to the age of the fruits. The concentration of JAs and ACC was higher in the 30-day-old than in the 15-day-old fruit, while an opposite trend was observed for the CKs, SA, putrescine, and spermidine (Figure 4a, Table 5). Regarding the hormonal treatments, GA4/7 increased the concentrations of SA but decreased the JAs and CKs contents as compared to intrinsic parthenocarpy (Figure 4a, Table 5). GA4/7 and GA3 treatments had opposite effects on the total GA content in 15-day-old fruit: GA3 increased it while GA4/7 decreased it compared to intrinsic parthenocarpy. However, this difference was mainly observed for GA19 but not for the bioactive GAs. GA4/7 also increased putrescine content in 15-day-old fruit compared to intrinsic parthenocarpy. For its part, 6BA increased GA content in 30-day-old-fruit and increased spermidine and putrescine content as compared to intrinsic parthenocarpy (Figure 4a, Table 5).

**Table 5.** Statistical results (ANOVA2) of endogenous phytohormone profile of early fruit development of intrinsic and extrinsic parthenocarpic 'Conference' pear fruits.

| Parameter | Treatments | Fruit Stages | Interaction |
|---|---|---|---|
| ABA | $F_{4,12}$ = 0.58, $P = 0.6808$ | $F_{1,12}$ = 0.98, $P = 0.3412$ | $F_{4,12}$ = 0.44, $P = 0.7777$ |
| IAA | $F_{4,12}$ = 2.40, $P = 0.1076$ | $F_{1,12}$ = 1.27, $P = 0.2810$ | $F_{4,12}$ = 1.03, $P = 0.4300$ |
| SA | $F_{4,12}$ = 8.08, $P = 0.0021$ | $F_{1,12}$ = 50.73, $P < 0.0001$ | $F_{4,12}$ = 4.54, $P = 0.0182$ |
| BzA | $F_{4,12}$ = 2.36, $P = 0.1118$ | $F_{1,12}$ = 1.22, $P = 0.2914$ | $F_{4,12}$ = 1.18, $P = 0.3673$ |
| JAs | $F_{4,12}$ = 9.03, $P = 0.0013$ | $F_{1,12}$ = 285.97, $P < 0.0001$ | $F_{4,12}$ = 9.23, $P = 0.0012$ |
| GAs | $F_{4,12}$ = 17.84, $P < 0.0001$ | $F_{1,12}$ = 16.18, $P = 0.0017$ | $F_{4,12}$ = 16.84, $P < 0.0001$ |
| bioactive CKs | $F_{4,12}$ = 0.35, $P = 0.8365$ | $F_{1,12}$ = 16.26, $P = 0.0017$ | $F_{4,12}$ = 0.54, $P = 0.7078$ |
| CK-*N*-glucosides | $F_{4,12}$ = 1.18, $P = 0.3684$ | $F_{1,12}$ = 24.24, $P = 0.0004$ | $F_{4,12}$ = 2.96, $P = 0.0649$ |
| CK-*O*-glucosides | $F_{4,12}$ = 0.10, $P = 0.9799$ | $F_{1,12}$ = 41.97, $P < 0.0001$ | $F_{4,12}$ = 0.34, $P = 0.8471$ |
| CK phosphates | $F_{4,12}$ = 1.03, $P = 0.4319$ | $F_{1,12}$ = 49.30, $P < 0.0001$ | $F_{4,12}$ = 1.29, $P = 0.3334$ |
| ACC | $F_{4,12}$ = 2.00, $P = 0.1593$ | $F_{1,12}$ = 52.22, $P < 0.0001$ | $F_{4,12}$ = 3.40, $P = 0.0445$ |
| spermidine | $F_{4,12}$ = 3.18, $P = 0.0356$ | $F_{1,12}$ = 54.63, $P < 0.0001$ | $F_{4,12}$ = 2.34, $P = 0.0900$ |
| spermine | $F_{4,12}$ = 1.37, $P = 0.2813$ | $F_{1,12}$ = 17.31, $P = 0.0005$ | $F_{4,12}$ = 0.92, $P = 0.4732$ |
| putrescine | $F_{4,77}$ = 1.37, $P = 0.0073$ | $F_{1,12}$ = 134.09, $P < 0.0001$ | $F_{4,12}$ = 5.18, $P = 0.0050$ |

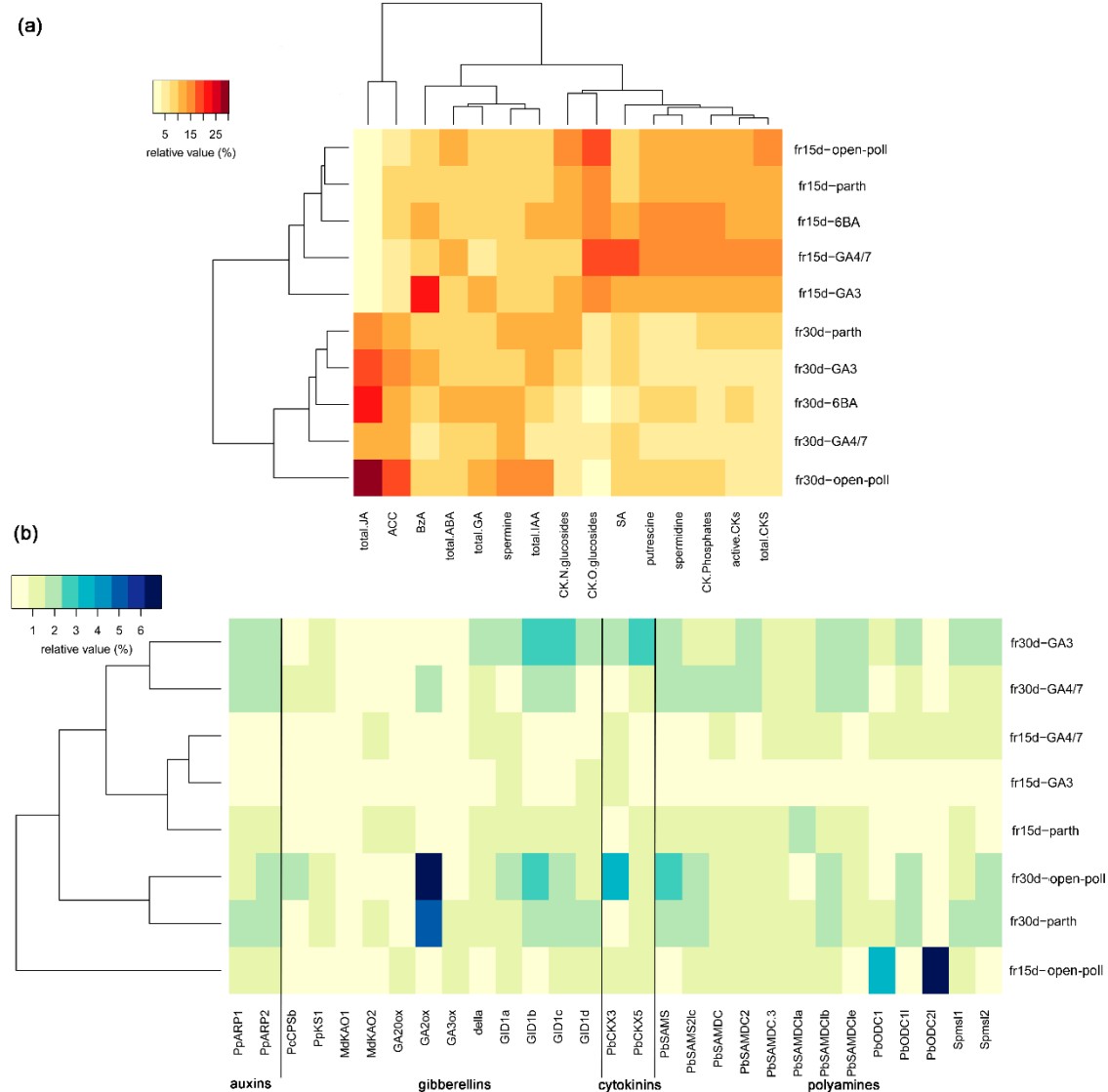

**Figure 4.** Effects of extrinsic parthenocarpy on hormonal regulation during early fruit development of 'Conference' pear fruits. (**a**) Heatmap of phytohormone and polyamine profiles in young 'Conference' fruits after different hormonal treatments; relative distribution expressed in %. (**b**) Heatmap of relative transcript levels of genes involved in phytohormone and polyamine metabolism and signaling in 'Conference' young fruits after different hormonal treatments. Fifteen-day-old fruit (fr15d) and 30-day-old fruit (fr30d) after gibberellin (GA3, GA4/7) and cytokinin (6BA) treatments, self-pollination (parth), and open-pollination (open-poll). Phytohormone abbreviations are as described in Figures 1 and 2; gene annotations are as described in Table 1.

### 3.2.3. Effects of Extrinsic Parthenocarpy on Expression of Genes Involved in Phytohormone Metabolism

Expression of genes involved in phytohormone and PA metabolism and signaling differed between extrinsic and intrinsic parthenocarpy (Figure 4b). Gene expression also varied with the fruit developmental stage mainly in extrinsic parthenocarpic fruits. Indeed, the expression of genes involved in auxin, GA, CK, and PA metabolism or signaling were globally less expressed in the 15-day-old than in the 30-day-old GA-treated fruits after treatment (Figure 4b). The differences between fruit stages were less pronounced for the open-pollinated and intrinsic parthenocarpic fruits. Compared to intrinsic parthenocarpy, extrinsic parthenocarpy decreased the expression of *ARP* genes (*PpARP1* and *PpARP2*) at 15 days but not 30 days after treatment. The expression of some genes

involved in GA metabolism were also decreased by extrinsic parthenocarpy compared to intrinsic parthenocarpy as observed for *GA20ox* and *MdKAO2* (only for GA3) in 15-day-old fruits or *GA2ox*, *GA3ox* and *MdKAO2* in 30-day-old fruits. However, the effects of extrinsic parthenocarpy on genes involved in the GA response and CK catabolism depended on the fruit developmental stage and the applied phytohormone. Transcript levels of genes involved in GA response were mainly decreased 15 days after treatment with GA3 (*DELLA, GID1b, GID1c*) or GA4/7 (*GID1b, GID1c, GID1d*), while 30 days after treatment, they were mainly increased in response to GA3 (*DELLA, GID1a, GID1b, GID1c*) and decreased in response to GA4/7 (*GID1a, GID1d*) as compared to intrinsic parthenocarpy. Regarding CK catabolism, GA3 increased the expression of *PbCKX3* whatever the fruit stage while expression of *PbCKX5* was decreased by GA3 and GA4/7 in 15-day-old fruits and increased by GA3 in 30-day-old fruits compared to intrinsic parthenocarpy. Genes involved in PA metabolism were also affected by extrinsic parthenocarpy. Most of them showed a lower transcript level 15 days after treatment with GA3 (*PbSAMS, PbSAMS2lc, PbSAMDC, PbSAMSDC2, PbSAMDC3, PbSAMDClb, PbSAMSDCle, Spmsl1*) or GA4/7 (*PbSAMS, PbSAMS2lc, PbSAMSDC2, PbSAMSDCle*) compared to intrinsic parthenocarpy. However, 30 days after treatment, expression of most PA genes was either increased (*PbSAMDC2, PbSAMSDCle* for GA3 and *PbSAMDC, PbSAMDC2, PbSAMDCl2* for GA4/7) or decreased (*PbSAMS2lc* for GA3 and *PbODC1, PbODC1, Spmsl1, Spmsl2 for* GA4/7) compared to intrinsic parthenocarpy.

## 4. Discussion

Although the transcriptomic, proteomic and metabolomic aspects of fruit development have begun to be investigated in pear [5,6,14], the involvement of phytohormones and PAs in fruit set and early fruit development remains unclear. Taking advantage of the ability of the *Pyrus communis* cv. 'Conference' to produce intrinsic parthenocarpic fruits, we compared the phytohormone and PA metabolism in fertilized and intrinsic parthenocarpic fruits. Our results confirm that intrinsic parthenocarpy occurs in 'Conference' and that parthenocarpic fruits are more elongated and have a smaller size than pollinated fruits [16]. Fruit size is indeed correlated with seed development in pear [20,37], and it is well known that seeds communicate through phytohormones to the surrounding tissues to promote fruit growth [7,11]. We indeed observed differences in phytohormone and PA profiles between flowers resulting from fertilization and intrinsic parthenocarpy 3 days after treatment. However, the phytohormone and PA profiles of open-pollinated flowers were more similar to the profiles of parthenocarpic flowers than to the profiles of cross-pollinated flowers. Such a difference could be explained by the low number of normal seeds that developed in open-pollinated fruits as compared to cross-pollinated fruits. The lack of normal seed development may affect phytohormone metabolism during early fruit development. The size of open-pollinated fruits was nevertheless similar to the size of cross-pollinated fruits, and aborted seeds were observed in open-pollinated fruits suggesting that fertilization took place in open-pollinated flowers but that seed development was aborted. It should be noted that self-incompatibility symptoms (limited pollen tube growth and large callose plugs) were observed in 38% of the open-pollinated flowers while none was observed in cross-pollinated flowers (data not shown). The presence of incompatible pollen on the stigma and arrest of pollen tube growth may affect the phytohormone profile and explain the observed differences between open-pollinated and cross-pollinated flowers. It would be interesting to investigate the hormonal metabolism separately in seeds and pericarp during fruit development to better understand their interaction in the regulation of pear fruit set and growth. It is nevertheless known that the number of normal seeds in 'Conference' fruits is at least two or three times lower than in other pear cultivars even when flowers are hand-pollinated with compatible pollen [16,20,38]. Our results also showed that differences of phytohormone and PA concentrations between fertilization and intrinsic parthenocarpy were more evident in flowers 3 days after treatment than in 15- and 30-day-old fruits. In the same way, phytohormone and PA profiles clearly differed between flowers and young fruits. This observation is not restricted to the phytohormones as Oikawa et al. also observed large differences for all metabolites between flowers at anthesis and young fruits (15- and 30-day-old) [6].

The main differences observed between fertilization and intrinsic parthenocarpy were an increase in IAA, ABA, ACC, and spermine and a decrease of SA, JAs, CK phosphates and putrescine in flowers 3 days after treatment. An increase in auxins after fertilization is consistent with their key role in fruit initiation [7,8,13]. However, IAA decreased later during fruit development as also observed in the pear cv. 'La France' [6]. The increase in IAA after fertilization was associated with a lower expression of the *ARP* genes in the fertilized flowers compared to intrinsic parthenocarpy. More surprising is the observed increase in ABA and ACC since ABA and ethylene usually occur in high concentrations in the flowers before pollination and subsequently decrease after pollination and fertilization [7,12]. Although we harvested the flowers 3 days after pollination, we may not exclude the fact that it was before the decrease of ABA and ACC concentrations since their levels were indeed low in 15- and 30-day-old fruits. A decrease of ABA concentration after anthesis was also observed in other pear cultivars [6]. The role of PAs, SA and JAs in fruit development is not yet well understood but they are required for fruit development, at least in some species [10,39]. Metabolomic analysis in pear showed that SA and JA levels were high in flower buds and decreased at fruit set while PAs increased during early fruit development [6]. Our results suggested that in 'Conference' pear, fertilization was associated with a decrease of putrescine and an increase of spermine. Such a modification was corroborated by the expression of genes coding for SAMDC and SPMS that were less expressed in the fertilized flowers compared to intrinsic parthenocarpy. We did not observe strong differences in the GA concentrations between fertilization and intrinsic parthenocarpy in 'Conference' despite the key role of these phytohormones in fruit development [4,7,13]. The high level of GAs in non-fertilized flowers could partly explain the intrinsic parthenocarpy observed in 'Conference'. Indeed, in the non-parthenocarpic European pear cultivar 'Starkrimson', an increase of GAs was observed after fertilization in 5-day-old fruits compared to non-pollinated flowers [21]. This result suggests that an increase of GAs was required to set fruit in this cultivar as non-pollinated flowers did not set fruit. However, genes involved in GA metabolism were expressed differently in fertilized and intrinsic parthenocarpic fruits in our study. Genes involved in GA synthesis (*GA20ox, GA3ox*) were more expressed in fertilized fruits or flowers while genes involved in GA degradation (*GA2ox*) were more expressed in intrinsic parthenocarpic fruits. An upregulation of *GA20ox* genes and a downregulation of *GA2ox* genes were also observed after fertilization in 'Starkrimson'. We also observed a downregulation of the gibberellin receptor (*GID1*) genes in parthenocarpic fruits. GID1 promotes the degradation of the DELLA proteins which negatively regulate GA signaling [40]. Expression of *DELLA* was not affected by fertilization and intrinsic parthenocarpy but we may not exclude a regulation at the protein level. Regarding CKs, we did not observe a modification in the bioactive CKs but rather a decrease of CK phosphate level and a decrease in expression of genes involved in CKs catabolism (*CKX*) in response to fertilization. Cytokinins are usually increased during fruit set and play a key role in fruit development [7,12]. However, in their study, Oikawa et al. observed a decrease of the most active CKs in young pear fruits as compared to flower buds [6].

Although 'Conference' could set intrinsic parthenocarpic fruits, extrinsic parthenocarpy induction by spraying phytohormones is a common practice in 'Conference' orchards [16,24,41]. We compared intrinsic and extrinsic parthenocarpy on fruit set and production as well as on phytohormone and PA metabolism. Hormonal treatments did not improve fruit set or fruit size as compared to intrinsic parthenocarpy in our experiments. However, previous studies have reported that GA treatment could stimulate parthenocarpic fruit production in 'Conference' under adverse growing conditions such as frosts [16,24]. Treatment with GAs, CKs or melatonin were reported to induce extrinsic parthenocarpic fruits in several European and Asian pear cultivars [4,22,23,39]. Moreover, GA treatments were also reported to increase fruit size in European and Asian pears [4,14,23,42]. We observed that GA and CK treatments affected the phytohormone and PA profiles and metabolism and that the effects varied according to the hormone tested. Total GA content was decreased by GA4/7 while it was increased by GA3 in 15-day-old fruit compared to intrinsic parthenocarpy. GA3 and GA4/7 also have different efficiency in extrinsic parthenocarpy induction in Asian pear [4]. Only GA4/7 induced parthenocarpic

development in *Pyrus bretschneideri* and such extrinsic parthenocarpy induction was associated with an increase in GA level in young fruit, although GA4/7 treatment did not strongly affect the expression of genes involved in GA metabolism in this species [4]. The authors explained this discrepancy by the high level of exogenous GA contribution so that endogenous synthesis was no more necessary to induce fruit development [4]. A similar hypothesis could explain that genes involved in GA metabolism were globally decreased by GA3 and GA4/7 applications in our experiments. In *Pyrus bretschneideri,* GA4/7 negatively regulated *GID1* and *DELLA* expression in 14-day-old fruits [4]. We also observed a lower expression of those genes in response to GA3 and GA4/7 applications in 15 day-old fruits while some of these genes were later upregulated in 30-day-old fruits. However, when 'Starkrimson' pear flowers were treated with melatonin, which has a similar function as IAA, extrinsic parthenocarpy was induced by promoting GA biosynthesis at both transcriptomic and metabolomic levels [21]. Those results suggested that GAs play an important role in parthenocarpy in pear [4,6,21]. We may hypothesize that in 'Conference', intrinsic parthenocarpy occurs because the level of GAs is already high in non-fertilized flowers and that exogenous application of GAs is thus not required to induce fruit development. It would be necessary to compare phytohormone metabolism during early fruit development in several parthenocarpic and non-parthenocarpic pear cultivars to check if this hypothesis may explain the differences between intrinsic and extrinsic parthenocarpy in European pear. However, GAs alone are not sufficient to induce parthenocarpy in 'Conference' as previously reported [16]. Extrinsic parthenocarpy induced by hormonal treatments also affects other phytohormone levels and gene expressions. For example, in *Pyrus bretschneideri,* GA4/7 increased IAA level by upregulation of genes coding for auxin efflux carrier [4] and GAs downregulated the *ARP* genes [43]. We did not observe an IAA increase in response to GA treatments but a decrease of *ARP* expression. Our results also suggested a role for SA, JAs and PAs in extrinsic parthenocarpy. Further studies are required to unravel the hormone interactions in fertilized and parthenocarpic fruit development in pear.

## 5. Conclusions

In summary, our study showed that intrinsic parthenocarpy occured in the European pear cv. 'Conference', and that phytohormone and PA profiles and metabolism differ in response to fertilization and intrinsic parthenocarpy processes in the flowers 3 days after treatment. An increase in auxins, abscisic acid, ethylene precursor, and spermine and a decrease in putrescine were observed in the fertilized flowers as compared to the intrinsic parthenocarpic ones. No strong differences were observed between fertilization and intrinsic parthenocarpy in 15- and 30-day-old fruits. Fertilization also upregulated genes involved in GA synthesis and down-regulated genes involved in GA catabolism, although the total GA content was not modified. Such differences in phytohormone and PA metabolism could be related to seed development and explain the fruit shape differences between fertilized and parthenocarpic fruits. However, extrinsic parthenocarpy induction by exogenous GA and CK applications did not increase fruit set and fruit size as observed in other pear cultivars. Exogenous GA application did not strongly modify the endogenous GA level while affecting the expression of genes involved in GA metabolism. We hypothesize that the intrinsic parthenocarpy of 'Conference' could be related to the high GA level in the flowers, explaining why exogenous GA application did not increase parthenocarpy as observed in other pear species and cultivars.

**Author Contributions:** Conceptualization, M.Q. and A.-L.J.; methodology, M.Q., C.B., V.M., P.I.D.; formal analysis, M.Q.; investigation, M.Q., C.B., V.M., P.I.D.; resources, M.Q., A.-L.J.; data curation, M.Q., C.B.; writing—original draft preparation, M.Q.; writing—review and editing, M.Q., A.-L.J., V.M..; visualization, M.Q.; supervision, A.-L.J.; project administration, A.-L.J.; funding acquisition, M.Q. and A.-L.J.

**Funding:** This research was funded by Service Public de Wallonie (SPW), convention number D31-1260, D31-1296, and D31-1337. Part of this research (PID, VM) was supported by the Czech Science Foundation (grant no. 16-14649S) and by the Ministry of Education, Youth and Sports of CR from European Regional Development Fund-Project "Centre for Experimental Plant Biology" (no. CZ.02.1.01/0.0/0.0/16_019/0000738)

**Acknowledgments:** The authors are grateful to the Centre Fruitier Wallon (CEF) for access to their orchards. The authors thank T. Van der Veken, M. Warzée, and T. Mabeluanga for field observations.

**Conflicts of Interest:** The authors declare no conflict of interest. The funders had no role in the design of the study; in the collection, analyses, or interpretation of data; in the writing of the manuscript, or in the decision to publish the results.

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
