# Peer review of "Hormonal Regulation of Early Fruit Development in European Pear (Pyrus communis cv. ‘Conference’)"

_horticulturae, doi:10.3390/horticulturae5010009_

Round 1

Reviewer 1 Report

Please look at the notes within the file

Author Response

The response to the reviewers could be found in the attached file

Reviewer 2 Report

The manuscript titled ”Hormonal regulation of early fruit development in European pear (Pyrus communis cv. ‘Conference’)” deals with the investigation of phyto-hormones involved in the growth process of parthenocarpic and non-parthenocarpic fruits of Conference pear. Althought the advancement of the state of the art provided by this work is limited, it deserve publication on this journal after minor revisions

Minor comments

Pants were thinned? They had all the same fruit set at harvest…

L 115 Replace “was putted” with was placed

L381-383 actually the fruit size was similar between control and cross-pollinated plant while the number of well formed seeds was quite smaller in control trees. This point deserve deeper discussion taking into account other hypothesis. There was a different initial fruit set but the lower rate recorded in cross pollinated treatment should indicate a lower competition during the first stage of fruit growth. There could have been a sort of stenospermocarpy in control trees? Why? Please improve the discussion of this point

Author Response

Response to the reviewer comments could be found in the attached file
